# Recombinant Lactaptin Induces Immunogenic Cell Death and Creates an Antitumor Vaccination Effect in Vivo with Enhancement by an IDO Inhibitor

**DOI:** 10.3390/molecules25122804

**Published:** 2020-06-17

**Authors:** Olga Troitskaya, Mikhail Varlamov, Anna Nushtaeva, Vladimir Richter, Olga Koval

**Affiliations:** Institute of Chemical Biology and Fundamental Medicine SB RAS, 630090 Novosibirsk, Russia; troitskaya_olga@bk.ru (O.T.); mvarlamov@gmail.com (M.V.); nushtaeva.anna@gmail.com (A.N.); richter@niboch.nsc.ru (V.R.)

**Keywords:** milk peptides, recombinant lactaptin, immunogenic cell death, antitumor vaccination, indoleamine 2,3-dioxygenase inhibitor

## Abstract

Natural compounds of various origins are intensively investigated for their antitumor activity. Potential benefits of antitumor therapy can be achieved when cytotoxic agents kill cancer cells and these dying cancer cells drive adoptive immunity to the tumor. This strategy was successfully demonstrated for chemotherapeutic drugs that induce immunogenic type of cell death (ICD) with release of DAMPs (danger associated molecular patterns) and exposure of “eat me” signals. In this study, we demonstrated that recombinant human milk peptide lactaptin (RL2) induces death of cancer cells with ICD hallmarks in vitro with the release of ATP and high-mobility group box 1 protein (HMGB1) and exposure of calreticulin and HSP70 on the external cell membrane. RL2-treated cancer cells were efficiently engulfed by phagocytic cells. Using the syngeneic mouse model, we demonstrated that RL2-treated MX-7 rhabdomyosarcoma cells confer long-term immune-mediated protection against challenge with live MX-7 cells. We also analyzed the combinatorial antitumor effect of vaccination with RL2-treated cells and the inhibition of indoleamine 2,3-dioxygenase (IDO) with ethyl pyruvate. Compared to solo anti-tumor immunization with RL2-treated cells, additional chemical inhibition of IDO demonstrated better long-term antitumor responses than vaccination alone.

## 1. Introduction

Human milk is rich in various bioactive proteins and peptides, which regulate functions of the intestinal tract, vasculature, nervous system, immune system, and endocrine system [1]. Resulting from the cytotoxic activity of milk fraction against various cancer cells in vitro, the proteolytic fragment of milk′s k-Casein-lactaptin was purified and identified [2]. Recombinant analogue of lactaptin (RL2), covering the 57–134 amino acid sequence of k-Casein, was also efficient against tumors grafted onto mice [3,4,5,6]. The comparison of RL2 primary structure and the mechanism of its penetration into the cell suggests that it can be assigned to the cell penetrating peptide (Appendix A) [7,8]. The N-terminal region, which is essential for proapoptotic activity, is more ordered than its C-terminal counterpart and contains a site with a propensity for α-helical secondary structure [9]. RL2 was shown to induce cell death with the hallmarks of various types of programmed cell death: apoptosis, ATP depletion-dependent necrosis, and autophagy [3,10,11].

Induction of cell death with special molecular patterns can make dying cells immunogenic. A special set of danger signals, DAMPs, exposed on the cell surface or secreted by cells after cell death induction are the main hallmarks of the immunogenic type of cell death (ICD). These molecular characteristics of ICD are surface-exposed calreticulin (CRT) and heat shock proteins 90 and 70 (HSP90 and HSP70) and extracellular ATP released high mobility group protein B1 (HMGB1). The Nomenclature Committee on Cell Death has recently defined ICD as a form of regulated cell death that is sufficient to activate an adaptive immune response in immunocompetent syngeneic hosts [12]. Nowadays, various clinically used anticancer drugs are demonstrated to induce ICD of cancer cells, but pathways leading to such outcomes are not fully understood [13,14,15,16].

“Find me” and “eat me” DAMPs of dying cells recruit dendritic cells (DCs) and stimulate their maturation. DCs take up tumor-associated antigens, followed by their presentation to T-cells with the activation of antitumor responses. To overcome immune system responses, tumors use various immune suppression or tolerance mechanisms. Expression of CD31 and CD47 “don’t eat me” traits prevent efficient uptake of tumor cells by antigen-presenting cells [17,18]. Secretion of immunosuppressive cytokines (TGFβ, IL-10) and metabolic enzymes by the tumor microenvironment is another obstacle for the immune system [19,20]. Immunosuppressive properties have been described for several metabolic enzymes such as inducible nitric oxide synthase (iNOS), indoleamine 2,3-dioxygenase (IDO), tryptophan 2,3 dioxygenase, arginase, and others [21,22]. Therefore, we supposed that the immunosuppressive molecules could be inhibited to reinforce antitumor vaccination. In addition, the widely used IDO inhibitor 1-methyl-DL-tryptophan (1-MT) and the low-cost, anti-inflammatory agent ethyl pyruvate (EP) have also shown a potent immune-based antitumor response [21,23,24].

Nowadays, only a restricted number of proteins or peptides have been shown to trigger ICD in cancer cells. Among them, peptide RT53 and RIG-I-like helicases (RLH) all induced the hallmarks of immunogenic cell death [25,26].

Here, we first examined whether recombinant peptide lactaptin RL2 stimulates ICD hallmarks in cancer cells in vitro. Second, using MX-7 mice rhabdomyosarcoma cells, we provide evidence that immunization of syngeneic mice with RL2-treated cells partly prevents the growth of tumors of the same type of cells. In addition, we investigated if the inhibition of immunosuppressive enzyme could increase the efficiency of antitumor immunization with RL2-treated cells.

## 2. Results and Discussion

Immunogenic cell death has in itself dual advantages for anti-cancer therapy: first, ICD inducers directly kill tumor cells, and second, dying cells can serve as a vaccine material that activate long-term antitumor response. Human adenocarcinoma MDA-MB-231 and MCF-7 cells and murine rhabdomyosarcoma MX-7 cells were used for our experiments. Previously, we had already calculated IC50 for lactaptin analogue RL2: for these cells, it was about 250–300 μg/mL [10,11]. This dose of RL2 was used for cell death induction in this study. We have previously shown that repetitive injections of RL2 (5–50 mg/kg) for 3–5 days effectively inhibited ascites and solid tumor transplant growth when administered intravenously or intraperitoneally, without obvious side effects [3]. Doxorubicin was the first chemotherapeutic drug described as an ICD inducer [13] with all the molecular hallmarks of ICD, so it was used as a positive control drug in our key experiments.

### 2.1. RL2 Induces Immunogenic Type of Cell Death (ICD) in Human and Murine Cancer Cells

Calreticulin translocation is one of the most essential markers of ICD. CRT movement from ER lumens to the plasma membrane is a rather early event and can be detected a few hours after chemotherapeutic treatment and before apoptosis-related phosphatidylserine exposure [27]. Phagocytic cells recognize ecto-CRT as “eat me” signals on cellular or apoptotic body membranes that have been swallowed up. The ecto-CRT level positively correlates with the immunogenic potential of apoptotic cells [28].

Cancer cells MDA-MB-231, MCF-7, and MX-7 were treated with RL2 at near IC50 concentration (0.3 mg/mL), and after incubation, the ecto-CRT was tested as well as the total cellular CRT (Figure 1). In addition, CRT mRNA was also analyzed. 

Flow cytometry revealed that after 4 h of incubation, more than 40 and 30 percent of the RL2-treated cells were ecto-CRT-positive in the MDA-MB-231 and MX-7 samples, respectively (Figure 1b). The increase of ecto-CRT-positive cells was time-dependent. MCF-7 cells were rather resistant to CRT translocation after RL2 and Dox treatment. The comparison of base CRT level in these cell lines showed its lower expression in MCF-7 cells (Figure 1c,d). To reveal whether ecto-CRT increased from its translocation or from the upregulation of CRT expression after treatment, we analyzed CRT mRNA and total CRT protein in the treated cells (Figure 1e–h). The analysis of total CRT did not reveal a positive regulation of this protein in RL2-treated cells. The CRT mRNA level of treated cells strongly correlated with total cellular CRT protein (Figure 1i–k). The decrease in CRT mRNA 5 h after the treatment led to a slight decrease in the CRT protein at 8 h of incubation (Figure 1g,h,i,k). Thus, the increase of ecto-CRT is a result of its RL2-stimulated translocation from the endoplasmic reticulum (ER). CRT-exposing dying cells can be recognized by dendritic cells (DCs) through the CD91 receptor followed by the antigen presentation and T-cell responses [29]. We suppose that MCF-7 cells with a low baseline CRT level (Figure 1c,d) can result in lower CRT translocation after an ICD inducer is applied, which can cause a weaker vaccination effect in vivo. Indeed, Obeid and co-authors have shown that apoptosis of cells with low baseline CRT is rather tolerogenic [30].

The release of HMGB1 from dying cells is a second hallmark of ICD. We observed that RL2 induced HMGB1 release to the culture medium at a high level after 12 h of incubation (Figure 2a,b). It was also confirmed by analysis of total cellular HMGB1 when we found a time-dependent decrease of cellular HMGB1, and it completely diminished by 24 h or 48 h of incubation with RL2 in the MX-7 cells and MDA-MB-231 cells, respectively (Figure 2c–f). Thus, we demonstrated that the decrease in cellular HMGB1 was due to its release from the treated cells. High HMGB1 release is preferable for ICD since low HMGB1 release or its low basal level in cancer cells is interconnected with a poor and insufficient activation of the TLR4 and RAGE receptors of immune cells [31].

ATP release in culture medium was assessed using a bioluminescent ENLITEN kit where luciferase converts luciferin using ATP, and a luminescent signal can be measured as described in the Methods. RL2 induces time-dependent ATP release from MDA-MB-231 and MX-7 cells. ATP released rapidly in MDA-MB-231 cells and it has already been well seen by 4 h of incubation. Moreover, by 24 h of incubation, a high level of ATP release was detected for both cell lines (Figure 2g,h).

Finally, we studied the RL2-dependent relocation of HSP70 to the outer cell membrane by flow cytometry. Ecto-HSP70-positive populations after 24 h were 66.2% for MDA-MB-231 and 56.2% for MX-7 cells (Figure 2i).

Thus, in cancer cells, RL2 activates the consensus set of hallmark contributors to the successful propagation of ICD.

### 2.2. Phagocytosis of RL2-Treated Cells

Since the efficient engulfment of dying cancer cells by DCs and by tissue residential macrophages is interrelated with subsequent long-term antitumor response, we analyzed if RL2-treated cells could be swallowed up by macrophages. For this goal, mice macrophages were investigated for their ability to engulf RL2-incubated dying cells or their corpses. Peritoneal macrophages of syngeneic C3H/He mice were isolated and labelled with green fluorescent cytoplasm dye. RL2-treated MX-7 cells were stained with non-toxic red fluorescent dye. Next, phagocytosis was assessed by the co-culturing of macrophages with RL2-treated MX-7 cells, followed by fluorescent microscopy. We observed a high level of phagocytosis of RL2-treated cells where the major part of red signals (MX-7 cells) were co-localized with green signals (macrophages). The relative phagocytosis rate was estimated by flow cytometry (Figure 3c,d). The double-positive green/red population was about 24% for the RL2-treated MX-7 cell specimens in contrast to 1% in the control specimens with intact MX-7 cells (Figure 3c,d).

### 2.3. Vaccination with RL2-Treated Cells

#### 2.3.1. Vaccination with the Whole Set of Treated Cells

To investigate whether transplantation of RL2-treated cells with activated DAMPs signals could initiate immune response against cancer, we performed a vaccination assay. Syngeneic C3H/He mice were immunized with MX-7 cells: the experimental group was subcutaneously injected into the left flank of mice with RL2-treated cells with no adjuvant and the control mice were injected with live cells (Figure 4a). Doxorubicin-treated cells (0.1 μg/mL, 24 h) were used as a positive control of ICD vaccination. We proposed that treated cells may cause antitumor immunity, which can be measured as the rejection of challenged viable cells. For these purposes, live MX-7 cells were transplanted into the opposite mouse flank. Tumor growth and mice survival was measured three times a week. We observed that this approach caused a vaccination effect in 50% of mice in the RL2 group and 50% of cases in the Dox group by the 70th day of immunization (Figure 4b). However, by this point in time, 30% of the mice had died in the RL2 group and all the mice were alive in the Dox group (Figure 4c). The mean tumor volume for the RL2 group was significantly smaller compared to that of the control group (Figure 4d). Our findings suggest that mice immunized by the RL2-treated MX-7 cells were partially protected against tumors of the same type of cells. Looking at the Dox and RL2 results, the last seemed to be weaker in antitumor vaccination. However, if we compare RL2 and another ICD-inducing protein—RIG-I-like helicase (RLH)—the vaccination effects are comparable; Duewell et al. demonstrated that vaccination of C57BL/6 mice with RLH-activated Panc02 tumor cells induced protective antitumor immunity in six out of eight mice [26]. 

Moreover, RL2-treated cells were more efficient in the prophylactic vaccination model than another peptide ICD inducer, RT53, which resulted in only drastically reduced tumor growth at the challenge site, with no tumor-free mice [25].

Together, our findings confirm the immunogenic properties of RL2-treated tumor cells in vivo.

#### 2.3.2. Vaccination Supplemented with Indoleamine 2, 3-dioxygenase (IDO) Inhibitor Treatment

We next studied whether the inhibition of indoleamine 2, 3-dioxygenase (IDO), the enzyme of tryptophan metabolism with immunosuppressive properties, could increase the vaccination effects of RL2-treated cells. To test our hypothesis, immunized mice were also treated with the IDO inhibitor ethyl pyruvate (EP) according to the scheme of Figure 5a. The rejection of a viable cell challenge was found in two of five mice in the RL2 group and in three of five in the RL2/EP group (Figure 5b). All mice were alive in the RL2/EP group, while in the RL2 group, one mouse died. In both experiments of vaccination, we observed dying mice in the RL2 group (Figure 4c and Figure 5c). It is likely that among the RL2-treated cells for vaccination, there was RL2-resistant live cells. The mean tumor volume for the RL2 and RL2/EP group was significantly smaller when compared to that of the control group (Figure 4d). It should also be noted that despite the fact that differences in the number of tumor-free mice for RL2 and RL2/EP were moderate, the average size of tumors for the RL2/EP group was the smallest and was about four-times lower than that for the control group (Figure 5d).

To verify the effect observed, we reduced the dose of cells for vaccination from 7 × 10^5^ to 5 × 10^5^. This strategy also led to an increase in the survival of the control mice and vaccinated mice (Figure 5e). The rate of the antitumor effect in the RL2 group was comparable to that in a previous experiment with a higher dose for vaccination: four out of 10 mice had no tumor in the challenge site and only one mouse died. In the RL2/EP group, six of 10 mice had no tumor and 10 out of 10 were alive. To summarize, according to the three independent experiments described above, an antitumor effect was observed for 43.3% of the vaccinated mice in the RL2 group. In the RL2/EP group, an antitumor effect was found for 60% of the mice (Figure 5e).

IDO is activated under various stress conditions including allograft rejection and tumor growth [32,33]. IDO catalyzes non-dietary tryptophan catabolism, and the breakdown of tryptophan in the tumor microenvironment and tumor-draining lymph nodes leads to DCs- and T-cell dependent immunosuppression [21]. Soliman and co-authors described that IDO inhibition can delay tumor growth and enhance dendritic cell vaccines [21]. Recently, Gao and co-authors have shown that IDO inhibition with specific small molecule NLG919 combined with doxorubicin exhibited better therapeutic effects on breast cancer in the murine 4T1 cancer model [34]. Despite the knowledge that the main IDO-producing cells are macrophages or dendritic cells in peripheral blood, other phenotypes of PBMCs have also been reported to express IDO: IDO mRNA was expressed in T-lymphocytes, B-lymphocytes, and natural killer (NK) cells [35]. This finding can partly explain how IDO plays a dual role in cancer. IDO production by plasmacytoid cells inhibits T-cell activation and proliferation with a cancer-promoting effect, while IDO in NK cells is essential to generate the killing activity against cancer cells [35]. More recently, Park and co-authors have shown that IDO suppressed NK cell function in thyroid cancer cells [36]. Moreover, IDO drives B-cell mediated autoimmunity with autoantibody production, which can be beneficial for anticancer immunity [37]. In summary, IDO plays a rather complex role in cancer and its activity in cancer progression has to be investigated in more detail. We suppose that the complexity of IDO as indicated above can explain our moderate success in increasing the vaccination effect with IDO inhibition.

#### 2.3.3. The Influence of IDO Inhibition on T- and B-Lymphocytes

To analyze the influence of IDO inhibition on T- and B-cells dynamic in mice blood, we used three time points: one day after the immunization of tumor cells, one day after the challenge of live MX-7 cells, and eight days after the challenge (Figure 6). We found that the challenge of live MX-7 cells led to the depletion of B-cells (CD3-/CD19 +) in mice blood eight days later in the RL2/EP experimental and control group compared to the level in healthy mice (Figure 6a). However, B-cell depletion in the RL2/EP group started earlier than in the control group, which was observed on the first day after the live MX-7 cell challenge. Thus, EP treatment stimulated B-cell depletion. At the same time, the T-cell population (CD3 +/CD19-) did not vary between the groups, but there was a difference between the investigated experimental or control group and healthy mice eight days after the live cells challenge (Figure 6b). The IDO pathway is also a major immunosuppressive pathway in immature myeloid-derived suppressor cells (MDSCs), which are represented by heterogeneous immunosuppressive cells in multiple cancer types [38,39]. It was shown that IDO is required for the immunosuppressive activity of MDSCs on T-cells, which can be blocked by the IDO inhibitor 1-methyl-L-tryptophan [40]. We suppose that the increase of CD3 +/CD19− cells is more the result of cell transplantation stimulus than IDO regulation. In this case, the total amount of T-cells does not directly reflect T cellular response to IDO activity. Overall, the indicated dualism of the IDO function has to be taken into account for further consideration of its role in drug-induced immunogenic cell death.

## 3. Materials and Methods 

### 3.1. Cell Lines and Mice

MX-7 murine rhabdomyosarcoma and MCF-7 human breast adenocarcinoma were obtained from the Russian Cell Culture Collection (Russian Branch of the ETCS, St. Petersburg, Russia). Human adenocarcinoma cells MDA-MB-231 (purchased: #ACC 732, DSMZ, Braunschweig, Germany) were maintained in Leibovitz L15 medium (Sigma-Aldrich) supplemented with 10% fetal bovine serum (GIBCO, Thermo Fisher Scientific, Waltham, MA, USA), 2 mM L-glutamine, 250 mg/mL amphotericin B, and 100 U/mL penicillin/streptomycin. MCF-7 human breast adenocarcinoma were grown in Iscove′s Modified Dulbecco′s Medium (IMDM, Sigma-Aldrich, St. Louis, MO, USA) supplemented with 10% fetal bovine serum (GIBCO, Thermo Fisher Scientific, Waltham, MA, USA), 2 mM L-glutamine, 250 mg/mL amphotericin B, and 100 U/mL penicillin/streptomycin. MX-7 cells were grown in Dulbecco’s Modified Eagle’s medium (DMEM, Sigma-Aldrich, St. Louis, MO, USA) supplemented with 10% fetal bovine serum (GIBCO, Thermo Fisher Scientific, Waltham, MA, USA), 2 mM L-glutamine, 250 mg/mL amphotericin B, and 100 U/mL penicillin/streptomycin. Cells were maintained as previously described [41]. 

Female C3H/He mice (6–8 weeks old) were obtained from the SPF vivarium of the Institute of Cytology and Genetics SB RAS, Novosibirsk, Russia.

### 3.2. Chemicals and Antibodies

RL2 was purified as described previously [42]. The following antibodies and chemicals were obtained from commercial sources: rabbit ant-h/m/rat CRT (Abcam, ab2907, Cambridge, UK), rabbit ant-h/m HMGB1 (Abcam, EPR3507, Cambridge, UK), goat anti-h/m/r GAPDH (R&D Systems, AF5718, Minneapolis, USA), mouse anti-human b-Tubulin (Sigma-Aldrich, T8328, St. Louis, MO, USA), mouse ant-h/rat HSP70 (R&D Systems, 841680, Minneapolis, USA), Alexa Fluor 594-conjugated chicken anti-rabbit antibodies (Invitrogen, Carlsbad, CA, USA), Alexa Fluor 555-conjugated goat anti-mouse antibodies (Invitrogen, Carlsbad, CA, USA), polyclonal rabbit-anti-mouse and mouse-anti-rabbit HRP-conjugated antibodies (Biosan, Novosibirsk, Russia), anti-mouse CD45 PerCP/Cy5.5 (Sony Biotechnologies, 1338530, San Jose, CA, USA), anti-mouse CD19 APC (Sony Biotechnologies, 1177650, San Jose, CA, USA), anti-mouse CD3 FITC (Sony Biotechnologies, 1101015, San Jose, CA, USA), anti-mouse CD25 PE (Sony Biotechnologies, 1110040, San Jose, CA, USA), and anti-mouse CD8a PE (R&D Systems, FAB116P, Minneapolis, USA).

Trypsin (Gibco, USA), inhibitor of trypsin from soybean (Paneco, Moscow, Russia), doxorubicin (Teva Pharmachemie B.V., Haarlem, Netherlands), and ethyl pyruvate (Sigma-Aldrich, St. Louis, MO, USA) were also used. 

### 3.3. Western Blotting 

Cells were lysed in buffer 20 mM Tris (pH 7.5), 1 mM EDTA, 150 mM NaCl containing 0.1% SDS (Sigma-Aldrich, St. Louis, MO, USA), 1% NP40 (Helicon, Moscow, Russia), 1% Triton X100 (Helicon, Moscow, Russia), 0.1% SDS, and 1x complete protease inhibitor cocktail (Roche Diagnostics GmbH, Mannheim, Germany). Lysates were centrifugated at 15,000 g for 20 min and protein samples (30 g) were separated by electrophoresis in SDS-polyacrylamide gel and transferred to a Trans-Blot nitrocellulose membrane (Bio-Rad, Hercules, CA, USA) by a wet blotting procedure (100 V, 500 mA, 90 min, 15 °C) using “Mighty Small Transphor” (GE healthcare Bio-Science AB, Buckinghamshire, UK). Immunodetection was performed by overnight incubation at 4 °C with one of the following antibodies: tubulin (1:200, Sigma-Aldrich, St. Louis, MO, USA), GAPDH (1:100, R&D Systems, Minneapolis, USA), calreticulin (1:1000, Abcam, Cambridge, UK), or HMGB1 (1:1000, Abcam, Cambridge, UK). After binding and washing, the membrane was incubated with goat anti-mouse HRP-conjugated polyclonal IgG (1:200, Abcam, Cambridge, UK) or donkey anti-goat HRP-conjugated IgG (1:200, R&D Systems, Minneapolis, USA) for 1 h and Novex ECL HRP chemiluminescent substrate reagent kit (Invitrogen, Carlsbad, CA, USA) for 1 min. A C-DiGit blot scanner (Li-COR Bioscience, Lincoln, NE, USA) was used for luminescent detection. Densitometric analysis of the western blot data was performed using the image analysis software Gel-Pro Analyzer (version 3.1. Medium Cybernetics, Rockville, MD, USA).

### 3.4. Flow Cytometry Analysis

All analyses were performed using a FACSCanto II flow cytometer (BD Biosciences, Franklin Lakes, NJ, USA), and the data were analyzed by FACSDiva Software (BD Biosciences). Cultured cells were initially gated (P1) based on forward scatter versus side scatter to exclude small debris, and ten thousand events from this population were collected. For ecto-CRT or ecto-HSP70 detection, cells were incubated with primary anti-CRT antibodies (1:100) or anti-HSP70 antibodies (1:100) for 1 h at 23 °C. Rabbit IgG (Thermo Fisher Scientific, Waltham, MA, USA) or mouse IgG (R&D Systems, Minneapolis, USA) was used as the isotype control.

From the whole, mice blood PBMC was isolated using Histopaque^®^1083 (Sigma-Aldrich, St. Louis, MO, USA). Next, cells were washed twice with PBS, followed by a 1 h incubation with anti-mouse CD45 PerCP/Cy5.5, anti-mouse CD19 APC, anti-mouse CD3 FITC, anti-mouse CD25 PE, and anti-mouse CD8a PE with a dilution of 1:1000. In flow cytometry analysis, samples were initially gated (P1) based on forward scatter versus side scatter to include lymphocytes. 

### 3.5. Quantitative Reverse Transcription Polymerase Chain Reaction (RT-PCR)

Total cellular RNA was isolated as described previously [43]. RT-PCR was performed in the one-tube reaction mixture BioMaster RT-PCR SYBR Blue (Biolabmix Ltd., Novosibirsk, Russia, www.biolabmix.ru) with gene-specific primers: calreticulin (CALR) 5′-GGGAACCCCCAGTGATTCAG-3′ and 5′-CCAGACTTGACCTGCCAGAG-3′; glyceraldehyde 6-phosphate dehydrogenase (GAPDH): 5′-GAAGATGGTGATGGGATTTC-3′ and 5′-GAAGGTGAAGGTCGGAGT-3′. To compare the PCR product yields, we performed real-time RT-PCR on a Bio-Rad iQ5 Cycler (Hercules, CA, USA). The relative quantification was given by the ratio between the mean value of the target gene and the mean value of the reference gene (GAPDH) for each sample. The mean Ct ratio (± SD) of three independent experiments is presented.

### 3.6. HMGB1 Assay

For the quantitative determination of HMGB1 in the culture medium, the enzyme immunoassay (IBL International, Hamburg, Germany) was used. Samples were prepared in order to the manufacturer’s protocols and their optical density was detected with a multichannel spectrophotometer Apollo LB912 (Berthold Technologies, Bad Wildbad, Germany) at 450 nm (reference wavelength 620 nm).

### 3.7. Adenosine Triphosphate Assay

Cells were seeded in 24-wells plates in IMDM medium supplemented with 10% FBS, 2 mM L-glutamine, and antibiotic/antimycotic solution with no Phenol red pH indicator. The next day, cells were treated with drugs for an indicated time and after that the culture medium was collected. Extracellular ATP was assessed in the medium using the ENLITEN ATP Assay System Bioluminescence Detection Kit (Promega, Madison, WI, USA), based on luciferin-luciferase conversion. For the measurement of the light intensity (560 nm), a luminometer CLARIOstar (BMG Labtech, Ortenberg, Germany) was used.

### 3.8. Ethic Statement

All animal experiments were carried out in compliance with the protocols and recommendations for the proper use and care of laboratory animals (EC Directive 86/609/EEC for animal experiments). The protocols were approved by the Committee on the Ethics of Animal Experiments of the Administration of the Siberian Branch of the Russian Academy of Science (Protocol Number 40, 04.04.2018). Mice were housed under super pathogen free (SPF) conditions in vented animal cabinets under controlled lighting conditions at 65% humidity, 25 °C, with 10/14 h light–dark cycle and allowed food and water ad libitum. Animals were euthanized by exposure to CO_2_.

### 3.9. In Vivo Vaccination Assay

MX-7 rhabdhomiosarcoma cells were seeded on a 25 mm^2^ culture flask, and cell death was induced in vitro by incubating the cells with RL2 (0.3–0.4 mg/mL) or Dox (0.1 μg/mL). After 48 h of incubation, cells were collected, washed once in PBS, and then resuspended in PBS. 

C3H/He mice were inoculated subcutaneously with 5 × 10^5^–7 × 10^5^ RL2-treated or Dox-treated MX-7 cells or live MX-7 cells (control group) on the left flank side. Seven days later, mice were challenged subcutaneously (s.c.) on the opposite flank with 2.5 × 10^5^–7 × 10^5^ live MX-7 cells. Tumor growth was evaluated using an electronic caliper every three days. 

### 3.10. Ethyl Pyruvate Treatment 

C3H/He mice were divided into three groups, and one of them was intraperitoneally injected with EP, 40 mg/kg (Sigma-Aldrich, St. Louis, MO, USA), four injections in total, every 12 h. For each experiment, a solution of ethyl pyruvate (2 mg/mL) was freshly prepared by dissolving the compound in Ringer’s lactate solution (Medpolymer, St. Petersburg, Russia). Control and RL2 groups were injected with solo Ringer’s lactate solution according to the same scheme.

### 3.11. Phagocytosis Assay

Macrophages were isolated as described previously [44]. Briefly, C3H/He were sacrificed, peritoneal macrophages were isolated from the lavage fluid of the peritoneal cavity with 3 mL of sterile 199 medium (GIBCO) supplemented with heparin (3 ME). Peritoneal fluid was collected into tubes gently and centrifuged at 150 g for 10 min. The pellet containing cells was resuspended in fresh 199 medium supplemented with 10% FBS, 2 mM L-glutamine, 250 mg/mL amphotericin B, and 100 U/mL penicillin/streptomycin. It was then seeded in four-well culture slides (BD Falcon, Bedford, MA, USA) and left for 1 h at 37 °C with 5% CO_2_. The non-adherent cells were removed by washing with fresh medium. The adherent cells were incubated in fresh 199 medium for 24 h at 37 °C with 5% CO_2_. After 24 h of incubation, macrophages were stained with 2 µM CellTracker™ Green CMFDA Dye (Thermo Fisher Scientific, Waltham, MA, USA) for 30 min. Then, the cells were washed with PBS.

Simultaneously, MX-7 cells growing in six-well culture plates were incubated with 2 µM CellTracker™ Red CMTPX Dye (Thermo Fisher Scientific, Waltham, MA, USA) for 30 min and washed with PBS. Then, MX-7 cells were harvested with trypsin, resuspended in 199 medium, and plated to a culture slide with stained macrophages. After 1 h of incubation, the non-adherent cells were removed by washing with PBS. Then, cells were fixed with ice cold methanol, covered with ProLong™ Gold Antifade Mountant (Thermo Fisher Scientific, Waltham, MA, USA). Stained cells were visualized using an Axioscop 2 PLUS fluorescence microscope (Carl Zeiss, GmbH, Jena, Germany).

### 3.12. Statistics‘

Significance was determined using a two-tailed, Student’s t-test using OriginPro 2015 software (Version 2015, OriginLab, Northampton, MA, USA). A significant difference resulted when the *p* value was less than 0.05. All error bars represent the standard error of the mean. In mice experiments, the differences in tumor-free mice between groups were calculated using non-parametric statistics, Pearson′s chi-square test, and these were significant with *p* < 0.05. 

## 4. Conclusions

Thus, we can conclude that recombinant milk peptide RL2 induces cell death in vitro with IDO hallmarks. In vivo, RL2-treated MX-7 rhabdomyosarcoma cells partly confer long-term immune-mediated protection against challenge with live MX-7 cells. 

The injections of ethyl pyruvate, an IDO inhibitor, increased the antitumor vaccination effect of RL2-treated cells. Moreover, injections of EP decreased the mortality of vaccinated mice. We suppose that this strategy of using EP can be expanded to other ICD inducers to increase their antitumor immunity. Further work is also necessary to determine whether other inhibitors of the tumor microenvironment’s immunosuppressive enzymes could enhance antitumor vaccination.

## Figures and Tables

**Figure 1 molecules-25-02804-f001:**
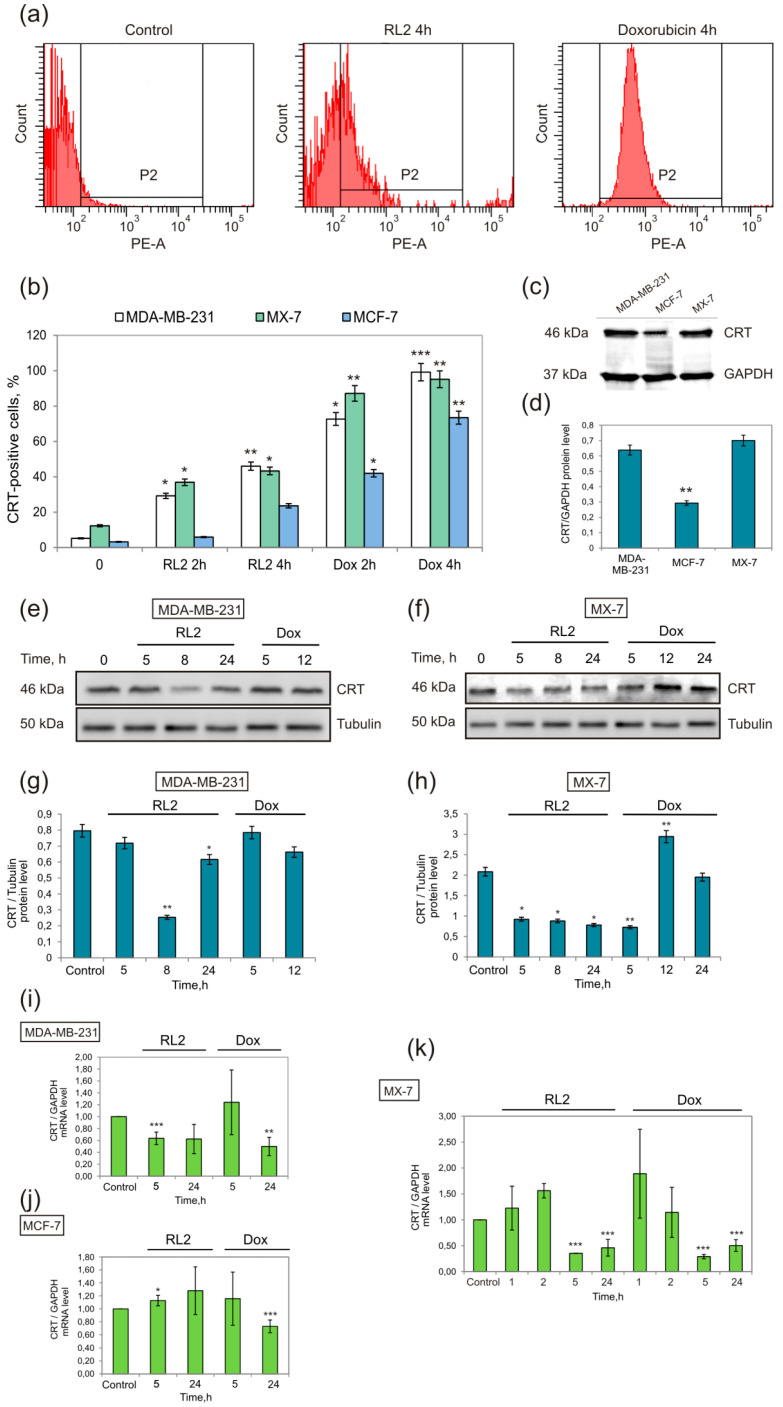
RL2 changes ecto-CRT in treated cells. Cells were treated with RL2 (0.3 mg/mL) or Doxorubicin (Dox, 0.1 μg/mL) for the indicated time. (**a**) Representative images of flow cytometry analysis of ecto-CRT (population P2) in MDA-MB-231 cells. (**b**) Relative amount of CRT-positive cells (population P2) according to flow cytometry data for MDA-MB-231, MCF-7, and MX-7 cell lines. (**c**,**d**) Western blot analysis of cellular CRT basal level (typical western and relative quantification of CRT/GAPDH); (**e**,**f**) Western blot analysis and (**g**,**h**) relative quantification of total cellular CRT in MDA-MB-231 cells or MX-7 cells treated with RL2 or Dox; (**h**–**k**) Expression levels of the CRT mRNA in RL2- or Dox-treated cells according to real-time polymerase chain reaction (PCR) analysis. Relative amount of CRT was normalized to GAPDH mRNA. Statistical differences between control and experimental groups are indicated by * for *p* < 0.05; ** for *p* < 0.01, *** for *p* < 0.001.

**Figure 2 molecules-25-02804-f002:**
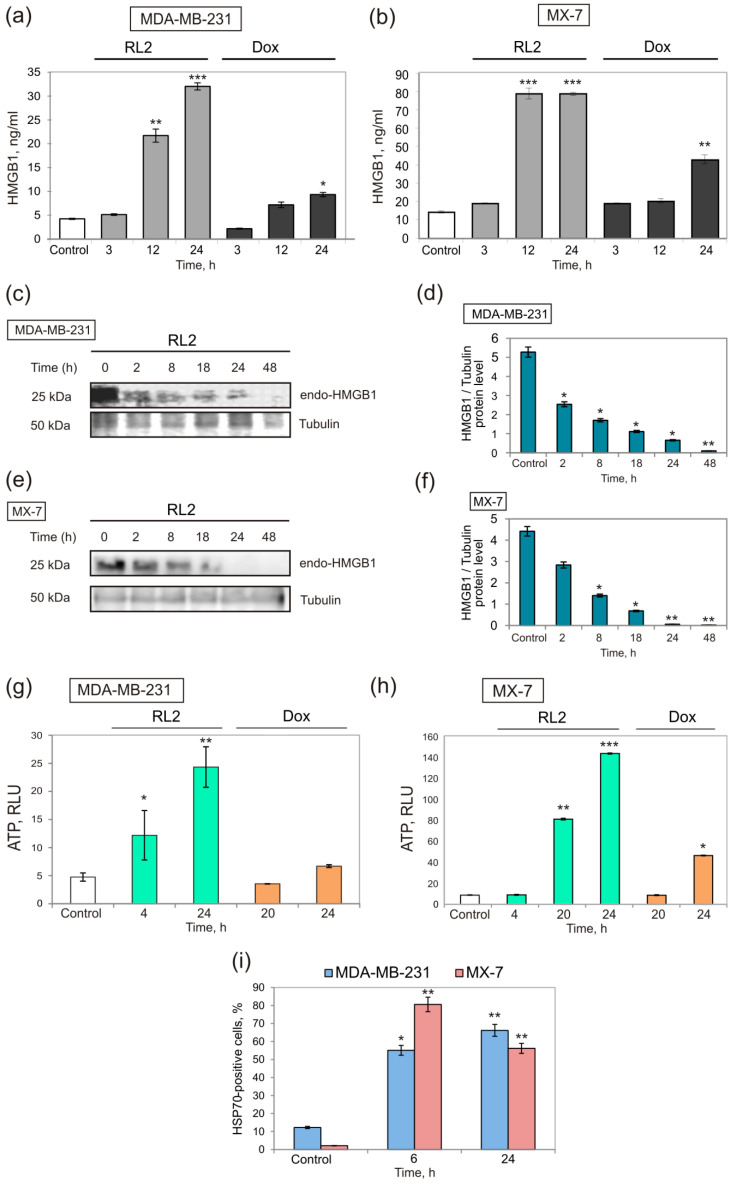
RL2 induces HMGB1 and ATP release and HSP70 translocation in treated cells. MX-7 and MDA-MB-231 cells were treated with RL2 (0.3 mg/mL) or Doxorubicin (0.1 μg/mL) for 2–48 h. (**a**,**b**) Extracellular HMGB1 in RL2- and Dox-treated cells; (**c**–**f**) Cellular HMGB1 in RL2-treated samples; western blot analysis of HMGB1 expression in cell lysates, one representative of two independent western blot experiments is shown and (**c**,**e**) relative quantification of HMGB1/Tubulin; (**g**,**h**) Relative amount of extracellular ATP, measured in cellular medium (RLU, relative luminescent units). (**i**) Surface-exposed HSP70 revealed by flow cytometry (RL2-treated cells). Median values of three independent experiments are shown ±SE. Statistical differences between control and experimental groups are indicated by * for *p* < 0.05; ** for *p* < 0.01, *** for *p* < 0.001.

**Figure 3 molecules-25-02804-f003:**
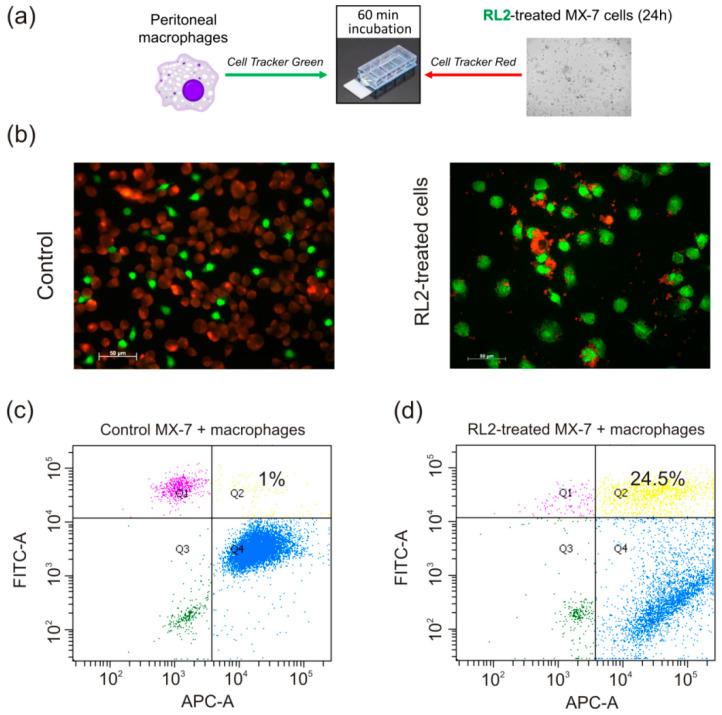
Phagocytosis of RL2-treated cells. (**a**) Scheme of the experiment. (**b**) MX-7 cells (red) were treated with RL2 (0.3 mg/mL) for 24 h and next co-incubated with peritoneal macrophages (green) for 60 min. Representative images of four independent experiments are presented. (**c**,**d**). Flow cytometry analysis of double-positive green/red population. Percent of double positive population is indicated in Quadrant 2 (Q2). Representative images are presented.

**Figure 4 molecules-25-02804-f004:**
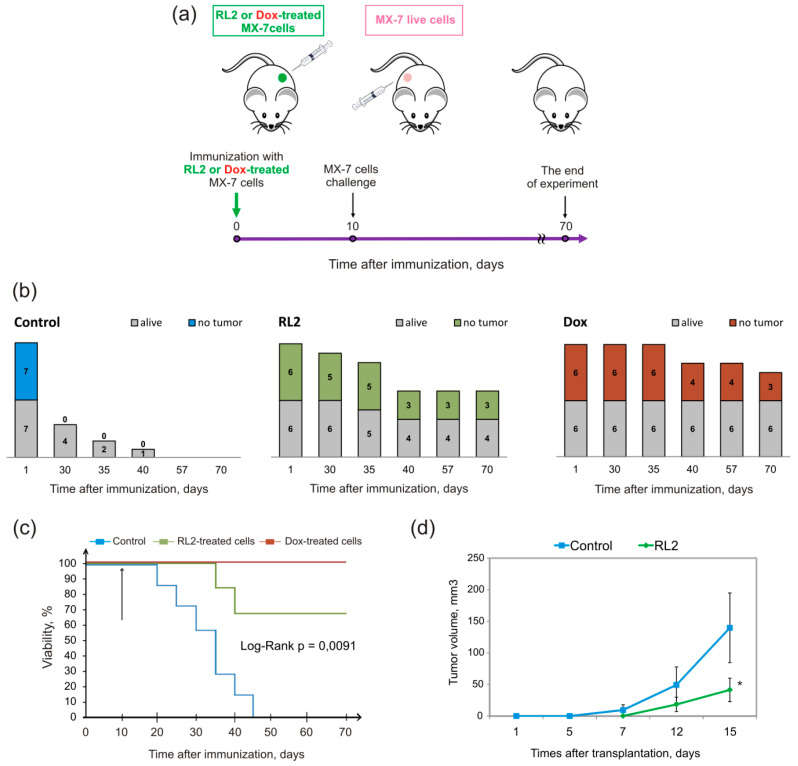
Immunization of C3H/He mice with MX-7 cells treated by recombinant lactaptin analog RL2. (**a**) Scheme of immunization. Syngeneic mice were s.c. immunized with RL2- or Dox-treated MX-7 cells (7 × 10^5^ cell/animal). Eight days later, the mice were challenged s.c. on the opposite flank with 7 × 10^5^ live MX-7 cells. (**b**) Dynamics of live and tumor-free mice in groups. (**c**) Viability of vaccinated mice. (**d**) Dynamic of tumor growth. The differences in tumor-free mice between groups were calculated using non-parametric statistics, the Pearson′s chi-square test, and these were significant with *p* < 0.05. Kaplan–Myer survival curves and Log-Rank statistics were calculated using GraphPad Prizm Software (San Diego, CA, USA).

**Figure 5 molecules-25-02804-f005:**
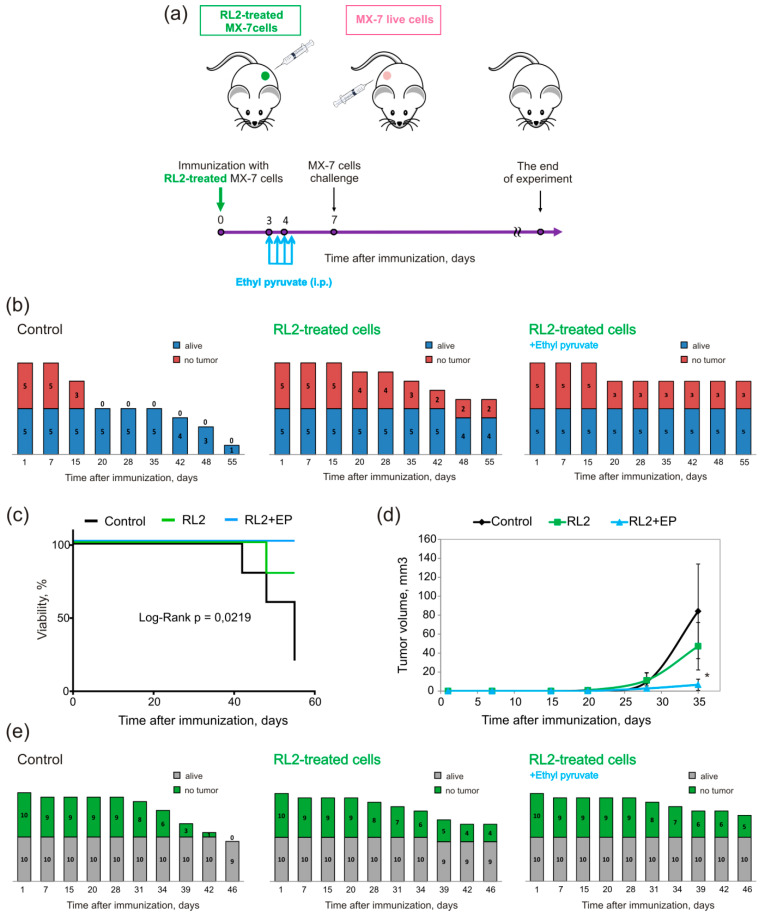
Ethyl pyruvate (EP) increases the vaccination effect of RL2-treated cells. (**a**) Scheme of C3H/He mice immunization with MX-7 cells treated by RL2 and EP injections. (**b**) Dynamic of live and tumor-free mice in groups. Syngeneic mice were s.c. immunized with RL2-treated MX-7 cells (7 × 10^5^ cell/animal) and three days after that mice were i.p. injected with EP (40 mg/kg) according the scheme. Seven days after, the mice were challenged s.c. on the opposite flank with 2.5 × 10^5^ live MX-7 cells. (**c**) Viability of vaccinated mice. (**d**) Dynamics of tumor growth. (**e**) Dynamic of live and tumor-free mice in groups. Syngeneic mice were s.c. immunized with RL2-treated MX-7 cells (5 × 10^5^ cell/animal), EP injected and MX-7 challenged according to the scheme. The differences between groups were calculated using non-parametric statistics, Pearson′s chi-square test, and these were significant with *p* < 0.05. Kaplan–Myer survival curves and Log-Rank statistics were calculated using GraphPad Prizm Software (San-Diego, CA, USA).

**Figure 6 molecules-25-02804-f006:**
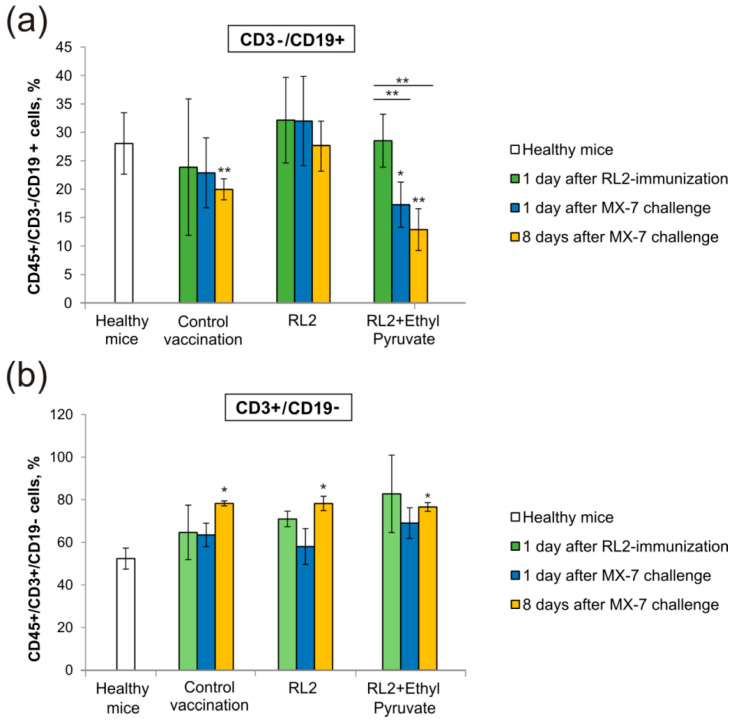
Distribution of immune cell populations in peripheral blood. PBMCs were harvested from whole blood samples, stained with CD45, CD3, and CD19 markers, and analyzed by flow cytometry. The number of positive cells is shown as a percentage of total blood CD45+ lymphocytes. (**a**) B-lymphocytes; (**b**) T-lymphocytes. Healthy mice were used as a negative control; “Control vaccination” mice were vaccinated with live MX-7 cells. Statistical differences between control and experimental groups are indicated by * for *p* < 0.05; ** for *p* < 0.01.

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
