# Peer review of "Recombinant Lactaptin Induces Immunogenic Cell Death and Creates an Antitumor Vaccination Effect in Vivo with Enhancement by an IDO Inhibitor"

_molecules, 2020, doi:10.3390/molecules25122804_

Round 1

Reviewer 1 Report

In the manuscript “Recombinant lactaptin induces immunogenic cell 2 death and creates an antitumor vaccination effect in 3 vivo with enhancement by an IDO inhibitor “ Koval and co-workers demonstrate that the recombinant human milk peptide lactaptin (RL2) is able to induce immunogenic type of cell death. In addition, the authors demonstrate by using syngeneic mouse model that RL2-treated MX-7 rhabdomyosarcoma cells confer long-term immune-mediated protection 18 against challenge with live MX-7 cells. Finally, the authors analyzed the combinatorial antitumor effect of 19 vaccination with RL2-treated cells and the inhibition of indoleamine 2,3-dioxygenase (IDO) with 20 ethyl pyruvate. Overall, the paper is very interesting and it is presented in a good shape. In my opinion, only a few minor points have to be addressed before publication.

  • The authors in the introduction should include a better description of the structural proprieties of the RL2 peptide. In addition, a picture of the peptide could be included in the supplementary materials or main text.
  • The authors have to correct several typos in the main text of the manuscript.
  • In figure 1 the panel letter (H) is reported two times. Moreover, I suggest to report the letter of each panel on the top of the figure.
  • The authors should report, to facilitate the readers, the same concentration unit when there is a comparison. For example the figure1 capture.

Author Response

Comment 1

In the manuscript “Recombinant lactaptin induces immunogenic cell 2 death and creates an antitumor vaccination effect in 3 vivo with enhancement by an IDO inhibitor “ Koval and co-workers demonstrate that the recombinant human milk peptide lactaptin (RL2) is able to induce immunogenic type of cell death. In addition, the authors demonstrate by using syngeneic mouse model that RL2-treated MX-7 rhabdomyosarcoma cells confer long-term immune-mediated protection 18 against challenge with live MX-7 cells. Finally, the authors analyzed the combinatorial antitumor effect of 19 vaccination with RL2-treated cells and the inhibition of indoleamine 2,3-dioxygenase (IDO) with 20 ethyl pyruvate. Overall, the paper is very interesting and it is presented in a good shape. In my opinion, only a few minor points have to be addressed before publication.

Answer 1

We thank the reviewer for the positive feedback on our manuscript.

Comment 2

            The authors in the introduction should include a better description of the structural proprieties of the RL2 peptide. In addition, a picture of the peptide could be included in the supplementary materials or main text.

Answer 2

In the current version of the manuscript we added description of the structural proprieties of the RL2 (See lines 32-35) as well as RL2 amino acids sequence in Supplementary 1.

Comment 2

  • The authors have to correct several typos in the main text of the manuscript.
    • In figure 1 the panel letter (H) is reported two times. Moreover, I suggest to report the letter of each panel on the top of the figure.
    • The authors should report, to facilitate the readers, the same concentration unit when there is a comparison. For example the figure1 capture.

Answer 3

  • We thank the reviewer for these comments. These typos were corrected.
  • In figure 1 the panel letter (H) was deleted and all the letters were moved to the top of the figure (See Fig. 1).
  • Dox concentration was transform to the “µg/ml” to uniform the units of concentration. (See Fig.1 and 2 captures and Lines 170, 366.

Reviewer 2 Report

In this study, Troitskaya et al., report the significant anti-tumor cytotoxic effects of recombinant human milk peptide lactaptin against human and murine tumor cell lines. Authors presented data in support of the activation of immunological cell death pathways in these tumor cells exposed to lactaptin in vitro. Further, they show that lactaptin exposed murine rhabdosarcoma tumor cells, when used as a cell vaccine afford long-term anti-tumor immunity and protection against re-challenge of live rhabdosarcoma cells in immunocompetent mice. Such tumor vaccine combined with additional targeting of major immunosuppressive pathway by using ethyl pyruvate to block IDO appears to enhance its effect in vivo.

For the most part, data presented are fairly convincing. The in vitro and in vivo experiments, the later in immunocompetent mice, included appropriate controls to draw conclusions.

One major concern is the grammatical errors throughout the manuscript that require extensive editing. Additionally the following concerns should be addressed:

  1. Figure 5. Kaplan-Myer survival curves should be shown and should be compared using a Log-Rank statistic.
  2. Figure 5 Legend: The cell numbers for (c) needs to be corrected.
  3. IDO pathway is also a major immunosuppressive pathway in immature myeloid suppressor cell population. Relevant literature should be cited and discussed.
  4. Line 239: Should be Figure 6.
  5. Figure 6: No statistics is shown; therefore, the data does not support the results presented in 2.3.3. Provide statistics and revise results accordingly.
  6. Figure 4 & 5: It is stated in methods that tumor growth was monitored using calipers. Tumor size data/tumor growth curves should be included. This will greatly strengthen the results.
  7. Panel (d) is missing in Figure 3 Legend. Please correct.
  8. The concentration of the drug used seems to be fairly high, in comparison to the positive control tested, and it is not clear whether such high concentrations are safe and feasible for in vivo tumor treatment. In fact, the potential of the drug for patient treatment has not been addressed.

Author Response

General Comments

In this study, Troitskaya et al., report the significant anti-tumor cytotoxic effects of recombinant human milk peptide lactaptin against human and murine tumor cell lines. Authors presented data in support of the activation of immunological cell death pathways in these tumor cells exposed to lactaptin in vitro. Further, they show that lactaptin exposed murine rhabdosarcoma tumor cells, when used as a cell vaccine afford long-term anti-tumor immunity and protection against re-challenge of live rhabdosarcoma cells in immunocompetent mice. Such tumor vaccine combined with additional targeting of major

immunosuppressive pathway by using ethyl pyruvate to block IDO appears to enhance its effect in vivo.

For the most part, data presented are fairly convincing. The in vitro and in vivo experiments, the later in immunocompetent mice, included appropriate controls to draw conclusions.

One major concern is the grammatical errors throughout the manuscript that require extensive editing.

Answer

We thank the reviewer for the high appreciation of our work. The extensive editing was performed.

Comment 1

Figure 5. Kaplan-Myer survival curves should be shown and should be compared using a Log-Rank statistic.

Answer 1

Kaplan-Myer survival curves was added to the Fig. 5 (see Fig. 5c) and compared using a Log-Rank statistic (as well as to Fig. 4c).

Comment 2

Figure 5 Legend: The cell numbers for (c) needs to be corrected.

Answer 2

The cell numbers were corrected. Please, see legend of Fig. 5.

Comment 3

IDO pathway is also a major immunosuppressive pathway in immature myeloid suppressor cell population. Relevant literature should be cited and discussed.

Answer 3

Relevant literature was cited and discussed (Please, see Lines 256-263).

Comments 4-7

  1. Line 239: Should be Figure 6.
  2. Figure 6: No statistics is shown; therefore, the data does not support the results presented in 2.3.3. Provide statistics and revise results accordingly.
  3. Figure 4 & 5: It is stated in methods that tumor growth was monitored using calipers. Tumor size data/tumor growth curves should be included. This will greatly strengthen the results.
  4. Panel (d) is missing in Figure 3 Legend. Please correct.

Answer 4-7

Line 239 (now line 265): was corrected. Figure 6 was updated with statistics, and results have been discussed (Please, see Figure 6 and Lines 249-256). Tumor growth curves were added for Fig. 4 and Fig. 5. (See Fig. 4d and Fig. 5 d) as well as they legends. Panel (d) was defined in Figure 3 Legend.

Comment 8

The concentration of the drug used seems to be fairly high, in comparison to the positive control tested, and it is not clear whether such high concentrations are safe and feasible for in vivo tumor treatment. In fact, the potential of the drug for patient treatment has not been addressed.

Answer 8

We added the information concerning the concentration of the RL2 used and its safety with the reference to our previous data (Please, see Lines 74-77).

Reviewer 3 Report

In this study the authors analysed the antitumor properties of recombinant human milk peptide lactaptin (RL2) on the human and murine cancer cells and the mechanisms that might be responsible for the increased immunogenicity of these cells. The subject of study is interesting and the data are obtained by adequate technology (flow cytometry analysis, Western blotting, RT-PCR, phagocytosis assay, in vivo vaccination assay). The study is well designed and the data provide information that might be helpful for the discovery of natural compounds with antitumor activity and new anti-tumor vaccine strategies. The manuscript deserves the publication, but I would recommend the corrections of points shown below.

Main findings of the study:

  1. Recombinant human milk peptide lactaptin (RL2) induces death of human and murine cancer cells in vitro with hallmarks of immunogenic cell death since it stimulates:
    1. release of ATP and high-mobility group box 1 protein (HMGB1)
    2. exposure of calreticulin and HSP70 on the external cell membrane
  2. RL2-treated cancer cells are efficiently engulfed by phagocytic cells
  3. immunization of syngeneic mice with RL2-treated cells partly prevents the growth of tumors of the same cell type
  4. vaccination with RL2-treated cells in combination with inhibitor of enzyme indoleamine 2,3-dioxygenase (IDO) leads to additional long-term antitumor responses in murine rhabdomyosarcoma MX-7 cells

Major revisions:

  • Lines 111-117 and lines 119-134 In the description of results please add the letters of graft about which you are talking in the text. At the present the reader is forced to find them himself in Figures 1 or 2.
  • Text on lines 111-117, which is related to data presented on Figure 1, should be placed before the Figure 1 (i.e. after line 91) and not bellow the Figure 2.
  • Similarly, it would be helpful if the text describing the data presented on Figure 2 (lines 119-134) are presented before the Figure 2.
  • The statistical significance of data presented on Figure1 (g-k) and Figure 5 (a, b) is missing.

Small mistakes and typos:

  • Line 13   DAMP only abbreviated, provide full name
  • Line 46 Check the grammar in sentence:   „maturation. DCs take up tumor-associated antigens, followed by they can present these to T cells“
  • Lines 49-51 The references are missing for the statement:  „Secretion of immunosuppressive cytokines (TGFβ, IL-10) and metabolic enzymes by tumor microenvironment is another obstacle for immune system. Immunosuppressive properties were described for several metabolic enzymes such as inducible nitric oxide synthase (iNOS), indoleamine 2,3-dioxygenase 52 (IDO), tryptophan 2,3 dioxygenase, arginase and others.
  • delete letter H on Figure 1
  • Line 120 Delete (?) in sentence ...“induced HMGB1 was released? to the culture medium at a high level after 12 hours of incubation“ and indicate in which cell line and under which letter this is shown on Figure 2. It is also unclear while on Fig. 2 e, f  the p value is shown only on data obtained after 24h and not also on those obtained after 12h.
  • Line 206  Correct tip-feller in   „after EP (40 mg/kg) were s.c. injectited according the scheme
  • Line 290 Use the same prefix for the expression of CRT and HSP on the external cell membrane in the text and in the methodology (ecto-CRT or exo-CRT?)

Author Response

General Comments

In this study the authors analysed the antitumor properties of recombinant human milk peptide lactaptin (RL2) on the human and murine cancer cells and the mechanisms that might be responsible for the increased immunogenicity of these cells. The subject of study is interesting and the data are obtained by adequate technology (flow cytometry analysis, Western blotting, RT-PCR, phagocytosis assay, in vivo vaccination assay). The study is well designed and the data provide information that might be helpful for the discovery of natural compounds with antitumor activity and new anti-tumor vaccine strategies. The manuscript deserves the publication, but I would recommend the corrections of points shown below.

Main findings of the study:

  1. Recombinant human milk peptide lactaptin (RL2) induces death of human and murine cancer cells in vitro with hallmarks of immunogenic cell death since it stimulates:
  2. release of ATP and high-mobility group box 1 protein (HMGB1)
  3. exposure of calreticulin and HSP70 on the external cell membrane
  4. RL2-treated cancer cells are efficiently engulfed by phagocytic cells
  5. immunization of syngeneic mice with RL2-treated cells partly prevents the growth of tumors of the same cell type
  6. vaccination with RL2-treated cells in combination with inhibitor of enzyme indoleamine 2,3-dioxygenase (IDO) leads to additional long-term antitumor responses in murine rhabdomyosarcoma MX-7 cells

Answer

We thank the reviewer for the positive feedback on our manuscript.

Comment 1

Lines 111-117 and lines 119-134 In the description of results please add the letters of graft about which you are talking in the text. At the present the reader is forced to find them himself in Figures 1 or 2.

Answer 1

In the description of results some references to the Fig. 1 and 2 were corrected according to the Reviewer’s suggestions (see Lines 90-103 and Lines 118-134).

Comments 2 and 3

Text on lines 111-117, which is related to data presented on Figure 1, should be placed before the Figure 1 (i.e. after line 91) and not bellow the Figure 2. Similarly, it would be helpful if the text describing the data presented on Figure 2 (lines 119-134) are presented before the Figure 2.

Answer 2 and 3

Text, which is related to data presented on Figure 1, was placed before the Figure 1 (i.e. after line 91) and not bellow the Figure 2. Similarly, the text describing the data presented on Figure 2 was moved before the Figure 2.

Comment 4

The statistical significance of data presented on Figure1 (g-k) and Figure 5 (a, b) is missing.

Answer 4

The statistical significance of data presented on Fig. 1 and Fig 2 was included. (We suppose that Reviewer means Fig. 2 when writes Fig.5 because the Fig.5(a) presentes a scheme of the experiment with no data).

Comments “Small mistakes and typos”

  1. Line 13 DAMP only abbreviated, provide full name

Answer

Full name of DAMP was provided (see Line 12).

Comment 2. Line 46 Check the grammar in sentence:   „maturation. DCs take up tumor-associated antigens, followed by they can present these to T cells“

Answer

Line 46 (now 48-49) was corrected.

Comment 3. Lines 49-51 The references are missing for the statement:  „Secretion of immunosuppressive cytokines (TGFβ, IL-10) and metabolic enzymes by tumor microenvironment is another obstacle for immune system. Immunosuppressive properties were described for several metabolic enzymes such as inducible nitric oxide synthase (iNOS), indoleamine 2,3-dioxygenase 52 (IDO), tryptophan 2,3 dioxygenase, arginase and others.

Answer

The references 19-22 were added (see Lines 53 and 55).

Comment 4. delete letter H on Figure 1

Answer

Extra “H” on Figure 1 was deleted.

Comment 5. Line 120 Delete (?) in sentence ...“induced HMGB1 was released? to the culture medium at a high level after 12 hours of incubation“ and indicate in which cell line and under which letter this is shown on Figure 2. It is also unclear while on Fig. 2 e, f  the p value is shown only on data obtained after 24h and not also on those obtained after 12h.

Answer

(?) was deleted. p value was also showed for 12h (see Fig. 2 a,b and others)

Comment 6. Line 206  Correct tip-feller in   „after EP (40 mg/kg) were s.c. injectited according the scheme

Answer

Line 206 was corrected (now 221-222).

Comment 7. Line 290 Use the same prefix for the expression of CRT and HSP on the external cell membrane in the text and in the methodology (ecto-CRT or exo-CRT?)

Answer

In the current version of the manuscript we use the same prefix for the expression of CRT and HSP – “ecto” (see Line 319).